# Lipid-driven Src self-association modulates its transformation capacity

Irrem-Laareb Mohammad[1,*], Marina I Giannotti[2,3,4,*] ✉, Elise Fourgous[5,6], Yvan Boublik[5,6], Alejandro Fernández[1] ✉, Anabel-Lise Le Roux[2] ✉, Audrey Sirvent[5,6] ✉, Marta Taulés[7] ✉, Serge Roche[5,6] ✉, Miquel Pons[1] ✉

Src tyrosine kinase regulates cell growth and adhesion through membrane signaling, and its deregulation is associated with cancer. Although active Src is anchored to the plasma membrane, the role of membrane lipids in its regulation remains unclear. Here, we report that Src self-associates via a lysine cluster in its SH4 region, a process mediated by lipids in human cells and in vitro. Mutation of the lysine cluster to arginine alters Src self-association and modulates its transforming function in human cells. Lipid-anchored micron-sized condensates of full-length Src form in supported homogeneous lipid bilayers (i.e., independently of lipid phase separation). Condensates also arise from the purified Src N-terminal regulatory element, which includes the myristoylated SH4 domain, the intrinsically disordered Unique domain, and the globular SH3 domain. However, the isolated SH4 domain alone forms small protein–lipid clusters rather than micron-sized condensates. Our findings reveal lipid-mediated kinase self-association as an additional regulatory mechanism for Src. This mechanism may also apply to other membrane-associated signaling proteins containing similar lysine clusters in their unstructured regions.

## Introduction

The non-receptor Src family (SFK) protein tyrosine kinases (TKs) are key regulators of membrane signaling that control cell growth and adhesion (Thomas & Brugge, 1997). Src, originally identified as an oncogene, is the prototypical SFK. It shares a common modular structure with other SFKs, consisting of an SH3, SH2, and a kinase domain (SH1) followed by a short C-terminal tail (Summy et al, 2003). In addition, Src contains an N-terminal unstructured region composed of a short myristoylated SH4 domain and a Unique domain (UD) with unclear function (Pérez et al, 2013). Deregulation of Src is frequently observed in human cancers and has been linked to tumor progression (Aligayer et al, 2002; Yeatman, 2004). However, SRC oncogenic mutations are rarely detected in human cancer, suggesting the existence of important non-genetic mechanisms driving Src kinase overactivation and cell transformation.

Crystallographic studies have identified a key conformational mechanism regulating Src catalysis, mediated by SH2- and SH3-dependent intramolecular interactions (Xu et al, 1999; Cowan-Jacob et al, 2005). However, this core regulatory mechanism alone may not fully account for the diversity of Src signaling and functions.

The myristoylated SH4 is a lipid-modified polybasic, intrinsically disordered region (IDR) essential for Src anchoring to the inner leaflet of the plasma membrane and for its transforming function. However, kinase regulation in this lipid context remains elusive. Membrane partition into cholesterol-enriched lipid rafts was reported to induce Src clustering with receptors, facilitating signaling (Mukherjee et al, 2003; Veracini et al, 2006, 2008; Holzer et al, 2011).

Small GTPases, extensively studied membrane-anchored signaling proteins, share with SFKs the presence of an analogous motif in their hypervariable region. This motif has been shown to regulate small GTPases via conformation-dependent lipid sorting and modulation of the membrane accessibility of other domains (Hutchins & Gorfe, 2024).

IDRs are often involved in the formation of phase-separated liquid protein condensates (Zhou et al, 2018; Hardenberg et al, 2020; Nesterov et al, 2021). Recent studies (Boyd-Shiwarski et al, 2022) and reviews (López-Palacios & Andersen, 2023; Gormal et al, 2024) highlight the potential role of liquid–liquid protein phase separation in kinase regulation. Src dimerization, involving its disordered N-terminal region, has been reported in mammalian cells and is associated with enhanced phosphorylation of specific substrates (Spassov et al, 2018).

The Src N-terminal IDR modulates its membrane anchoring, enabling membrane substrate phosphorylation and oncogenic

[1]Biomolecular NMR Laboratory, Department of Inorganic and Organic Chemistry, Universitat de Barcelona (UB), Barcelona, Spain   [2]Nanoprobes and Nanoswitches Group, Institute for Bioengineering of Catalonia (IBEC), The Barcelona Institute of Science and Technology (BIST), Barcelona, Spain   [3]Centro de Investigación Biomédica en Red de Enfermedades Raras (CIBERER), Madrid, Spain   [4]Materials Science and Physical Chemistry Department, IQTCUB, Universitat de Barcelona (UB), Barcelona, Spain   [5]CNRS UMR5237, University of Montpellier, CRBM, Montpellier, France   [6]Equipe Labellisée Ligue Contre le Cancer, CRBM, University of Montpellier, CNRS, Montpellier, France   [7]Centres Científics i Tecnològics (CCiTUB), Universitat de Barcelona (UB), Barcelona, Spain

Correspondence: Serge.Roche@crbm.cnrs.fr; mpons@ub.edu
*Irrem-Laareb Mohammad and Marina I Giannotti are first authors

signaling (Aponte et al, 2022). Mechanistically, the SH4 and UD of Src form a fuzzy complex condensed around the SH3 domain, maintaining extensive dynamics (Maffei et al, 2015; Arbesú et al, 2017) and modulating substrate selection and signaling. Unlike the short hypervariable region found in small GTPases, the SFK SH4 motif is directly linked to the Unique domain, forming a much longer IDR. Membrane anchoring by the lipidated SH4 domain may promote the formation of condensates involving the entire IDR, potentially via intermolecular interactions akin to those found in fuzzy complexes. Indeed, when anchored to membranes, the myristoylated disordered region of Src enables in vitro self-association, though the precise mechanism remains unclear (Le Roux et al, 2016a, 2016b). Despite being lipid-mediated, Src self-association does not require lipid phase separation, as it is observed in artificial membranes composed of a single lipid species.

Overall, these findings suggest an additional layer of Src regulation involving its unstructured region and membrane lipids. Here, we investigate the mechanism underlying crosstalk between Src's unstructured region and membrane lipids, focusing on lipid-mediated Src self-association (SA).

To this end, we used the following approach. First, using surface plasmon resonance (SPR) to detect SA, we identified SH4 residues essential for Src SA within the context of myristoylated SH4-UD-SH3 (referred to as the Src N-terminal regulatory element, SNRE) (Fig 1A). Next, we used atomic force microscopy (AFM) to characterize the nature of Src SA and to confirm its occurrence in vitro, including in full-length Src. We used neutral and anionic supported lipid bilayers that do not form lipid-separated phases to demonstrate that Src SA is intrinsic and independent of lipid phase separation. After identifying the essential role of a SH4 lysine cluster in Src SA, we showed that replacing these residues with arginine enhances SA. Translated these findings to human cells, we showed that lysine-to-arginine mutations in the SH4 domain similarly enhance Src SA in vivo and modulate Src's transforming capacity.

## Results

### The SH4 K5, K7, and K9 residues modulate lipid-mediated SNRE self-association

In a previous study, we designed an in vitro assay, shown schematically in Fig 1B, to assess Src binding kinetics to membrane lipids using SPR (Le Roux et al, 2016a). For this, we used immobilized lipid bilayers (SLBs) and purified SNRE from *E. coli* co-expressing yeast N-myristoyltransferase to induce protein myristoylation. Our SPR analysis revealed a predominant lipid-bound monomeric SNRE population, along with a minor form that remained persistently bound to the lipid membrane. Kinetic studies using SPR (Le Roux et al, 2016a), and single-molecule fluorescence (Le Roux et al, 2016b) indicated that the persistently bound form involves at least two SNRE molecules and requires the presence of lipids. To investigate the underlying mechanism, we focused on the contribution of the SH4 region through an alanine-scanning mutagenesis approach, targeting residues 3–10. We prioritized this region based on prior studies demonstrating that deletion of the first 10 residues

of the SH4 affected both the compaction of the fuzzy complex with SH3 (Arbesú et al, 2017) and the lipid membrane distribution and clustering (Dwivedi et al, 2017).

We first used SLBs composed of neutral zwitterionic lipid 1,2-dioleoyl-sn-glycero-3-phosphocholine (DOPC) in the liquid disordered ($L_d$) phase. The relative SPR responses of the individual mutants, compared with WT SNRE, showed that substituting lysine residues in positions 5 or 7 with alanine (K5A and K7A, respectively) reduced SA by ~30–40% (Fig 1C). Conversely, replacing the negatively charged aspartic acid at position 10 with alanine (D10A) increased SA. An increase in SA was also observed in the N4A mutant in neutral lipid conditions.

To assess the contribution of anionic lipids, we employed a 2:1 mixture of DOPC and 1,2-dioleoyl-sn-glycero-3-phospho(1'-rac glycerol) (DOPG), which does not form separate phases in our conditions (Himeno et al, 2014). This approach minimized potential complications arising from protein-induced lipid phase transitions while still enabling the comparison with zwitterionic lipids. Notably, the inhibitory effect of K5A and K7A mutations on SA was enhanced by ~10% in the presence of anionic lipids, and inhibition was also observed for the K9A mutant. These findings implicate an SH4 lysine cluster and negatively charged lipids for SNRE SA (Fig 1C). Furthermore, the S3A mutation reduced SA on anionic bilayers. This serine residue is positioned between the lysine cluster and the myristoylation site. We propose that it influences SA by stabilizing the orientation of positively charged residues when the myristoyl group is embedded in the lipid bilayer.

### K5A and K7A mutations hinder SNRE self-association independently of their effect on Src membrane affinity

Because SH4 K#A mutations may also affect the lipid binding affinity of individual Src molecules—and thus affect the actual concentration of Src on the membrane surface—we compared the SNRE SA capacity under conditions where the total density of bound proteins remained constant. These experimental conditions were achieved by varying the bulk concentration of injected SNRE variants (3–20 μM).

Under equal bound SNRE density conditions, SA formation was still reduced by ~40% for K7A and 50% for K5A compared with WT SNRE (Fig 1D). This finding indicates that the observed decrease in the SA population in the K#A mutants cannot be attributed solely to a reduction in protein density on the lipid surface. Interestingly, despite having the same overall charge, the steady-state response of the K5A and K7A mutants differed, suggesting that the specific positioning of the basic residues is important for primary binding and SA. We conclude that the lysine cluster in the SH4 drives SNRE SA in the presence of lipids, beyond its electrostatic contribution to binding negatively charged lipid membranes and despite the expected electrostatic repulsion between adjacent Src molecules.

### Src forms membrane-anchored condensates observed by atomic force microscopy on supported lipid bilayers

The requirement of a cluster of positive charged residues for Src SA on the membrane surface suggests that the negatively charged phosphate group of phospholipids could mediate interactions

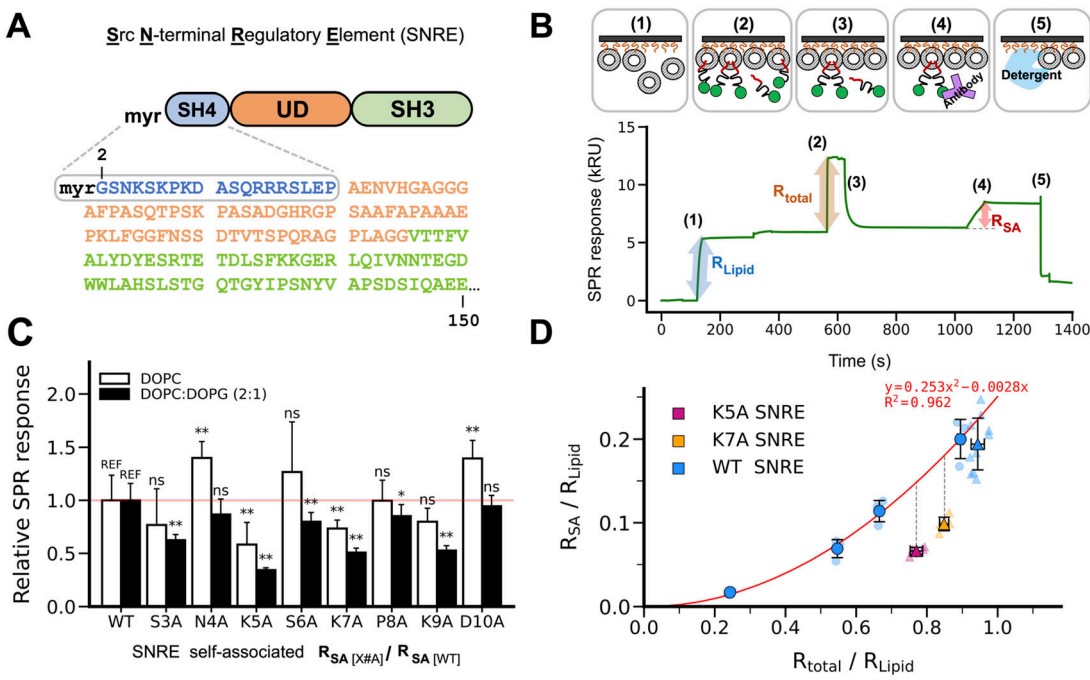

**Figure 1.  Lysine cluster in the SH4 domain participates in lipid-mediated Src N-terminal regulatory element (SNRE) dimerization.**
**(A)** Sequence of the SNRE. The SH4 and UD are intrinsically disordered. **(B)** Experimental protocol used to quantify self-association by surface plasmon resonance. Liposomes were immobilized using a phytosphingosine derivatized matrix. The total bound protein was quantified by the direct surface plasmon resonance response. After washing, only persistently bound species arising from self-association were retained and quantified by the response to an antibody not interfering with membrane binding. **(C)** Self-association quantification of SNRE mutants relative to the WT form using pure zwitterionic DOPC (open bars) or a mixture of DOPC with negatively charged DOPG (filled bars). Proteins were injected at 20 $\mu M$. Data are expressed as the mean ± SD, n = 4 (DOPC), n = 5 (DOPC: DOPG). Significant differences are indicated with asterisks (t test, *$P < 0.1$, **$P < 0.05$, ns, not significant; REF, reference). **(D)** Effect of K5 and K7 mutations on SA is not due to the reduced membrane affinity of the monomers. WT SNRE was injected at a bulk concentration of 3, 6, 10, and 20 $\mu M$, whereas the K#A mutants were injected at 20 $\mu M$. Data used originates from two different sensor chips indicated by the type of the marker (circle or triangle). Data are expressed as the mean ± SD, n = 3 (circles), n ≥ 5 (triangles).

between Src molecules. However, SPR does not provide information on the nature of the species formed by SA. Therefore, we investigated Src SA using AFM.

The addition of SNRE to a SLB composed of pure DOPC resulted in the formation of micron-sized condensates with a smooth surface about 1.5 nm thinner than the surrounding lipid bilayer (Fig 2A). Similar condensates were observed with myristoylated full-length Src (Fig 2B), although they exhibited distinct structural characteristics compared with those formed by the truncated protein. Specifically, the condensates formed by full-length Src had a thickness comparable to or exceeding that of the surrounding bilayer, which is consistent with the larger size of the protein (150 versus 536 residues for SNRE and full-length Src, respectively). In addition, the border region of the full-length Src condensate displayed a complex structure, in contrast to the smooth borders of the condensates formed by SNRE.

To further analyze lipid-mediated SA, we examined SLBs composed of a 4:1 ratio of DOPC:DOPG, which form homogeneous $L_d$ membranes. Indeed, in the absence of protein, homogeneous lipid bilayers were observed (Fig 3A). Upon the addition of SNRE, micron-sized condensates were also detected (Fig 3B and D).

The presence of negatively charged lipids resulted in notable differences in the appearance of the SNRE condensates compared to those formed with a single, zwitterionic, lipid species (DOPC). First, the condensates exhibited inhomogeneous thickness, with a height

profile fluctuating between 2 and 3 nm below the surrounding lipid bilayer. Force curve profiles within the condensate region confirmed this inhomogeneity but also revealed breakthrough events upon indentation, characteristic of lipid bilayers, suggesting the presence of lipids inside the protein-induced condensates (Fig S1). Second, the condensates formed in the presence of negatively charged lipids were surrounded by a "wall" protruding ~0.5–1 nm above the lipid surface (Fig 3D), a feature absent in DOPC-only bilayers.

In addition to the micron-sized condensates, numerous small nanometer-sized clusters (~0.5 nm in height, comparable to the surrounding "wall") were observed exclusively in the presence of negatively charged lipids. These findings suggest the protein preferentially interacts with negatively charged lipids, leading to the formation of smaller lipid–protein clusters that may remain isolated, accumulate at the borders of larger condensates to form a wall, or integrate into the condensates. This mechanism could explain the rougher surface of the condensates formed in DOPC: DOPG bilayers, compared with those formed in pure DOPC.

Similar features were observed with a larger construct incorporating the SH2 domain (Fig 3C), except for the thickness of the condensate, that is intermediate between that observed for the myr-SH4-UD-SH3 construct and full-length Src in DOPC. Unfortunately, full-length Src induced detachment of the DOPC:DOPG bilayer from the mica surface preventing further analysis of full-length Src in this system.

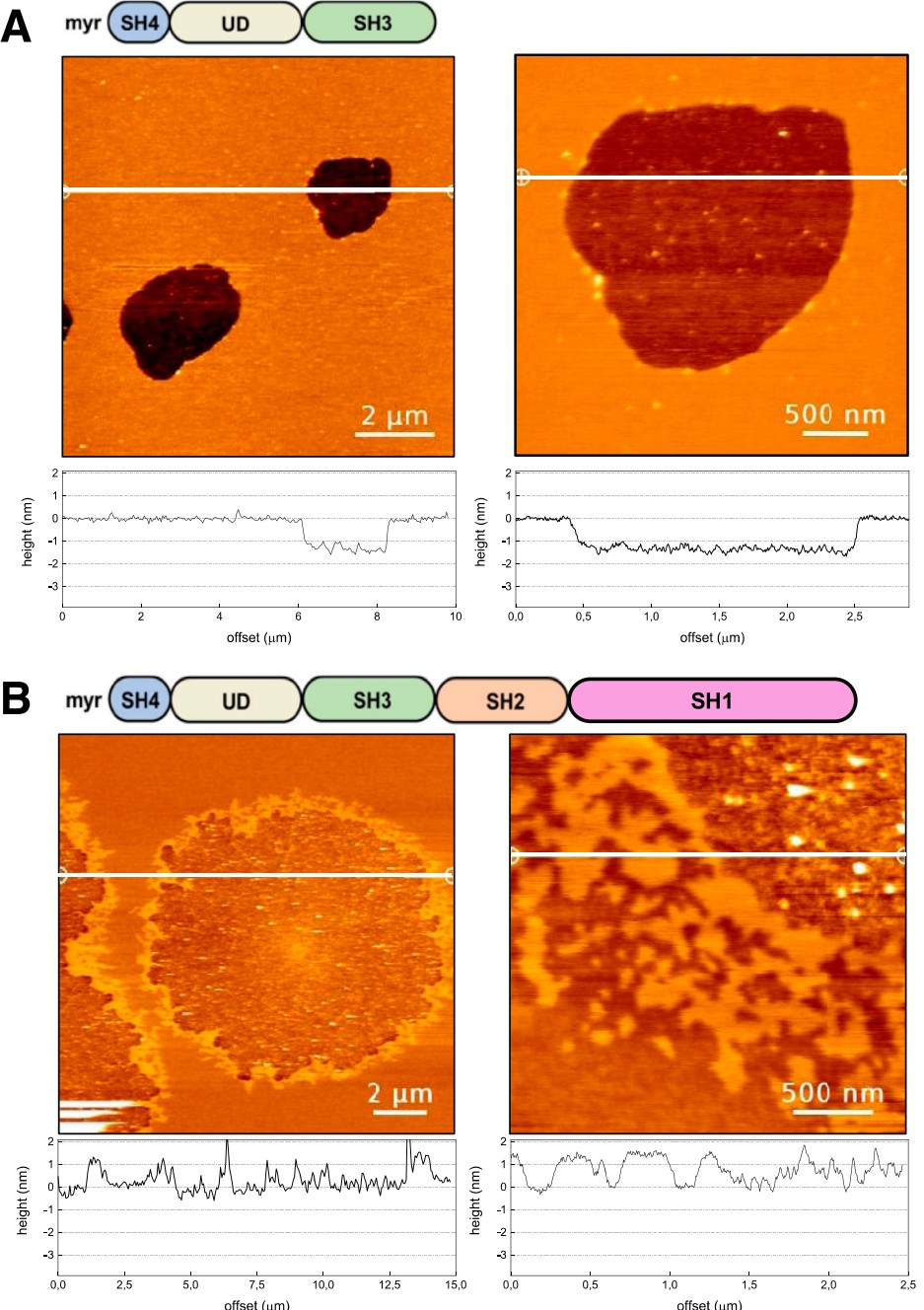

**Figure 2. Condensate formation by Src N-terminal regulatory element and full-length Src on supported pure DOPC lipid bilayers. (A, B)** Topographical atomic force microscopy images of (A) Src N-terminal regulatory element and (B) myristoylated full-length Src on mica-supported lipid bilayers made of pure DOPC. Cross sections along the indicated lines are shown underneath each image. The protein was added to the buffer solution covering the supported lipid bilayer, to a final concentration of 100 nM, and left to interact for 4 min before imaging. Images were acquired in working buffer (10 mM $NaH_2PO_4$/$Na_2HPO_4$, 150 mM NaCl, pH 7.5). Size bars are 2 $\mu$m (left) and 0,5 mm (right).

## Formation of small lipid–protein clusters is mediated by the SH4 domain, but micron-sized condensates require the presence of the Unique domain

If protein–lipid clusters are mediated by the lysine cluster, as suggested by the SPR experiments, then replacing lysine with arginine—known to exhibit stronger interactions with phosphate groups (Li et al, 2013; Neal et al, 2018; Thomas et al, 2024)—may alter the balance between discrete clusters and large condensates. To test this hypothesis, we generated a SNRE mutant in which K5, K7,

and K9 were replaced by arginine residues (3R mutant). Indeed, the 3R SNRE variant formed numerous small clusters but failed to generate micron-sized condensates (Fig 3E).

If the strength of the Src–lipid interaction influences condensate formation, then strong interactions between the basic residue cluster and a negatively charged lipid, such as phosphatidic acid (PA), might hinder condensate formation and instead promote discrete cluster formation. To investigate this, we tested WT SNRE in a mica-supported bilayer containing 10% dioleoyl phosphatidic acid (DOPA) and 90% DOPC. Similar to the behavior of 3R SNRE with

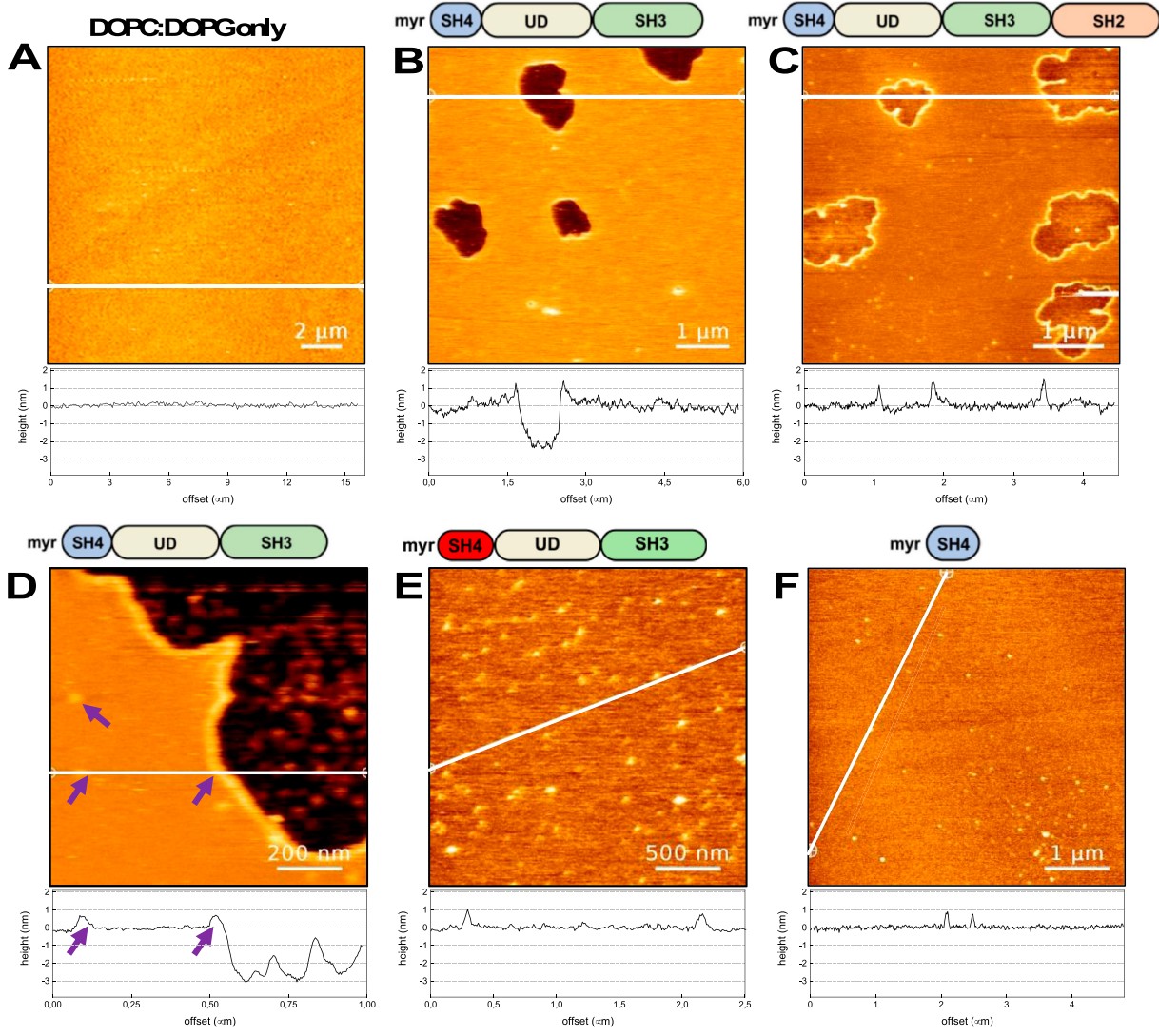

**Figure 3. Condensate formation by truncated Src forms in negatively charged DOPC: DOPG 4:1 supported lipid bilayers.**
Topographical atomic force microscopy images. **(A)** Homogeneous bilayer formed in the absence of protein confirms that DOPC: DOPG mixtures do not form separated lipid phases. The bar length is 2 $\mu$m. **(B)** Condensates formed by Src N-terminal regulatory element (SNRE). The bar length is 1 $\mu$m. **(C)** Condensates formed by a larger construct including the SH2 domain. The bar length is 1 $\mu$m. **(D)** Coexistence of micron- and nm-sized condensates formed by SNRE. Notice the similar height of the nm-sized clusters and the wall surrounding the micron-sized condensates. The bar length is 0.2 $\mu$m. **(E)** Only nm-sized clusters are formed in the 3R mutant of SNRE. The bar length is 0.5 $\mu$m. **(F)** Only nm-sized clusters are formed by the isolated myristoylated SH4 domain. The bar length is 1 $\mu$m.

DOPC-DOPG bilayers, WT SNRE in the presence of DOPA formed only discrete clusters (Fig S2), with no observable condensates. This finding supports the idea that the strength of the electrostatic interactions between acidic lipids and basic residue clusters regulates condensate formation. Interestingly, PA has been reported to bind and activate Src in response to sperm-induced phospholipase D activity during Xenopus egg's fertilization process (Bates et al, 2014).

Because condensate formation is often associated with intrinsically disordered regions, we examined the behavior of the myristoylated SH4 domain alone, which lacks the intrinsically disordered UD. The myristoylated SH4 domain formed only discrete clusters (Fig 3F), confirming that although Src SA involves the SH4 domain, the formation of large condensates requires additional regions.

Self-association, the role of the lysine cluster, and the enhanced self-association caused by the 3R mutation were confirmed in the isolated SH4 domain by SPR (Fig S3).

Overall, the AFM and SPR experiments support a lipid-mediated SNRE self-association mechanism, highlighting the critical role of the SH4 lysine cluster and anionic lipids in this molecular process.

## The SH4 lysine cluster regulates Src self-association in human cells

To investigate Src SA in human cells, we co-transfected in HEK293T cells with full-length Src constructs tagged at the C terminus with either Myc or a Flag sequence. Western blot analysis confirmed that these tags did not affect Src expression or activity,

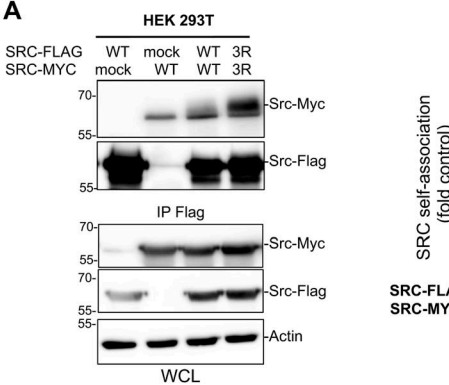

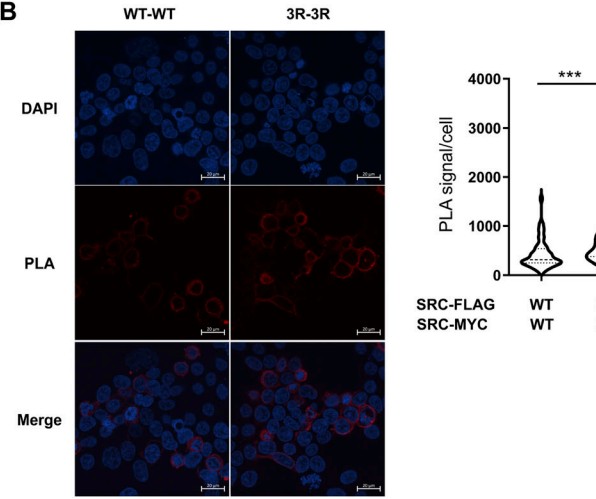

**Figure 4. Mutation of lysine to arginine in the SH4 lysine cluster enhances Src self-association in human cells.**
**(A)** HEK293T cells were transfected with the indicated Src constructs. Src-Flag proteins were immunoprecipitated (IP) from cell lysates and immunoblotted with the indicated antibodies. Immunoblots of whole-cell lysates were also performed as indicated. A representative example and the relative quantification of co-immunoprecipitation are shown. **(B)** Proximity ligation assay (PLA) from cells transfected with the indicated construct was used to prove the proximity of Src variants in intact cells. A representative example and the violin plot representation of the PLA signal quantification (PLA signal surface area/cell, six fields of 30–50 cells per experiment) as assessed by confocal microscopy are shown as the mean ± SEM; n = 3 independent experiments; ns, $P > 0.05$, $*P < 0.05$, $**P < 0.01$, $***P < 0.001$, $t$ test. Control experiments are presented in Fig S4 (expression and activity of tagged WT Src), Fig S5 (colocalization of WT and 3R Src), and Fig S6 (positive PLA control and whole-cell lysates).
Source data are available for this figure.

as assessed by the cellular phosphoprotein profile (Fig S4). Based on our in vitro SPR and AFM results showing enhanced SA in the 3R mutant and previous findings that lysine-to-arginine substitutions do not interfere with in vivo myristoylation (Kaplan et al, 1988), we compared SA of WT and 3R full-length Src (Src3R) in HEK293T cells. Confocal microscopy confirmed similar subcellular distribution patterns for the two Src variants, notably at the plasma membrane (Fig S5).

Using the tagged proteins, we detected Src SA via co-immunoprecipitation, demonstrating that Src3R exhibited a

50% increase in SA compared with WT Src (Fig 4A). SA was independently validated using a proximity ligation assay (PLA), which detects proteins located within ~40 nm of each other through immunofluorescence. Our approach was first validated by showing Src association with PEAK2, a well-established Src substrate and interactor (Fig S6) (Lecointre et al, 2018). Notably, Src3R exhibited a 100% increase in PLA-detected SA as compared to Src WT (Figs 4B and S6).

Overall, these data agree with our in vitro findings and confirm that Src undergoes SA in vivo, supporting our model of lipid-driven SA of membrane-anchored Src.

## Src self-association enhances oncogenic signaling in human cancer cells

To evaluate the functional relevance of Src SA, we assessed the impact of the Src3R mutants on oncogenic signaling. Src3R expression resulted in a 50% increase in cellular protein tyrosine phosphorylation, consistent with enhanced kinase activation, as indicated by elevated pY419 Src levels (Fig 5A). Next, we examined the effect of Src3R on Src transforming function. We previously demonstrated that Src overexpression enhanced the transforming potential of colorectal cancer cells, including the HCT116 cell line (Naudin et al, 2014; Sirvent et al, 2020). Using this model, we found that Src3R was more oncogenic than WT Src, as assessed on anchorage-independent growth and cell migration in the Boyden chamber (Fig 5B). We thus concluded that Src lipid-driven SA modulates Src signaling and oncogenic function, further supporting its role in cancer progression.

# Discussion

### A novel mechanism of Src regulation by membrane lipids

Our findings provide further evidence supporting a mechanism for Src regulation through the interaction with membrane lipids. We propose a model in which a cluster of positively charged lysine residues in the SH4 domain interacts with the negatively charged phosphate group of phospholipids to drive Src SA.

The functional importance of this SH4 lysine cluster was previously recognized in early studies (Kaplan et al, 1988; Silverman & Resh, 1992; Sigal et al, 1994). These studies established its role in in vivo myristoylation by N-myristoyltransferase and, more importantly, the importance of their electrostatic interaction with acidic lipids, which enhance the anchoring of individual Src molecules to the membrane. However, our study uncovers a previously unknown role of the SH4 lysine cluster as a key driver for lipid-mediated Src SA at the membrane surface.

We demonstrate that mutating the SH4 lysine cluster to arginine (Src3R mutant) enhanced Src SA in vitro and in vivo, leading to increased transforming capacity in HCT116 colorectal cancer cells. These findings suggest that modulating the strength of the protein–lipid interaction modulates Src SA, which in turn impacts cancer cell behavior.

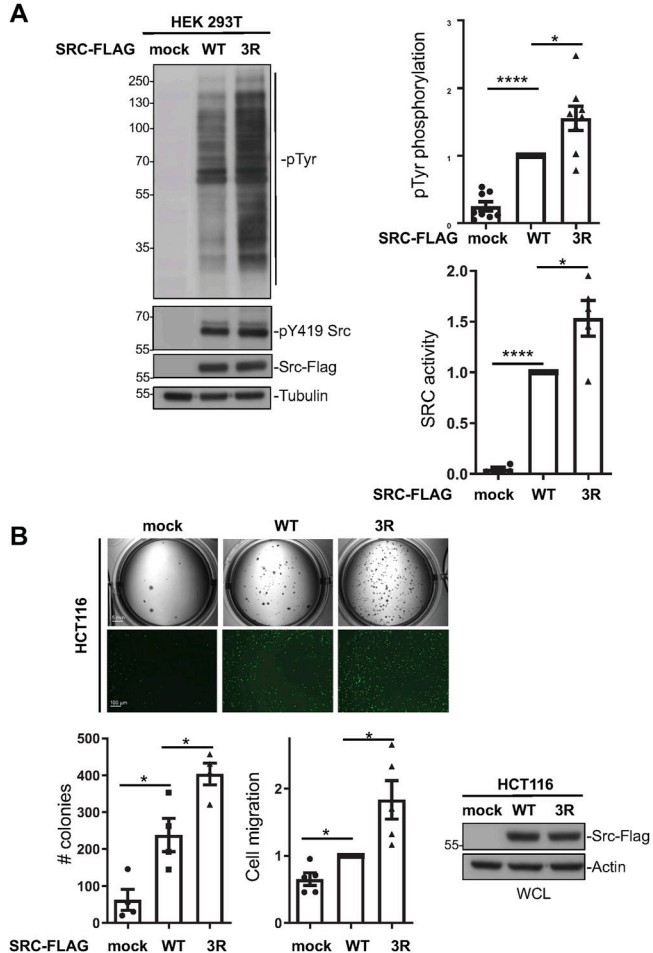

**Figure 5.  Src self-association increases oncogenic signaling in human cancer cells.**
**(A)** Phosphosignaling of the indicated c-Src variant transfected in HEK293T cells. A representative example and quantification of c-Src–induced protein tyrosine phosphorylation (pTyr) and Src activity (pY419 Src) assessed by immunoblotting. **(B)** Transforming activity of WT and 3R c-Src retrovirally transduced in a human colorectal cancer HCT116 cell line. An example (top panels) and quantification of c-Src–induced colony formation in soft agar and migration in the Boyden chambers. Size bars are 5 mm (colony formation) and 100 µm (migration assay). The level of transduced c-Src is also shown (bottom right). Data are shown as the mean ± SEM; n = 3–5; ns, $P > 0.05$, $*P < 0.05$, $**P < 0.01$, $***P < 0.001$, $t$ test.
Source data are available for this figure.

Previous research from our group indicated that the SH4 domain also participates in forming a fuzzy intramolecular complex, involving the Unique and SH3 domains. Fuzzy complexes arise from weak multivalent interactions that collectively produce strong macroscopic effects. Another well-documented consequence of multivalency is the formation of separate phases or condensates, often associated with disordered regions.

Using AFM, we observed micron-sized condensate formation by full-length Src and truncated variants in supported lipid membranes. These condensates are formed by lipid-mediated protein–protein interactions. In the presence of negatively charged lipids, smaller protein–lipid clusters are formed, coexisting with the large condensates. Presumably, the small clusters result from the recruitment of negatively charged lipids by a small number of protein molecules, whereas the larger condensates result from weaker, more delocalized interactions, and possibly direct protein–protein interactions away from the membrane surface. Interestingly, the isolated myristoylated SH4 domain and a longer construct with the 3R mutation form only small clusters. Small clusters, but no condensates, were observed in the presence of PA, although more extensive studies would be required given the complexity of phosphatidic acid–containing systems (Drabik & Czogalla, 2021). These observations suggest that lipid-mediated self-association is the first step of a more complex mechanism involving other Src regions.

The possible in vivo regulatory role of self-association, whether in the form of small clusters or condensates, remains an open question. Our in vivo data show that the 3R mutant has an enhanced transforming capacity. On the other hand, Src is known to be activated in vivo by PA (Stith, 2015). These two systems are shown to form only small clusters but no large condensates in the SLB. It is tempting to speculate that condensate formation may reduce Src activity, acting as a reservoir that can be activated through lipid remodeling, such as phospholipase-mediated changes in the lipid composition.

## Lysine clusters: a general mechanism for lipid-driven protein–protein interactions

Lysine clusters like those in Src are present in other SFKs (Silverman et al, 1993) including Fyn ([7]KDKEATK[13]), Yes ([5]KSKENK[10]), and Lyn ([5]KSKGKD[10]), or in K-Ras 2B ([180]KSKTK[184]). This suggests that a similar lipid-mediated regulatory mechanism may apply to other proteins containing lipidated polybasic domains.

Interestingly, the putative electrostatic contribution of basic residues to complement the weak interaction of a single myristoyl group in Src is not required in the other SFKs that have palmitoylated cysteines contributing to lipid binding. The well-recognized abundance of positively charged residues in the juxtamembrane region of integral membrane proteins is obviously not related to membrane affinity. A proposed role of the basic residue clusters is the selective recruitment of specific negatively charged lipids (Maxwell et al, 2018; Zhou et al, 2021; Araya & Gorfe, 2023). We propose that another key function of conserved lysine clusters is to drive lipid-mediated protein–protein association, thereby influencing membrane signaling pathways.

Lipid-driven protein–protein interactions represent an additional regulatory layer, exploiting the diverse lipid compositions found across cell types of cells or even among cells of the same type in different metabolic states. The concept of lipotypes—stable lipid configurations— associated with specific transcriptional programs and cell phenotypes (D'Angelo & La Manno, 2023) suggests that lipid-mediated Src SA may be cell type–specific. The finding that lipid-driven Src SA modulates Src oncogenic induction in tumor cells has obvious implications in cancer therapy. Aberrant lipid metabolism is a hallmark of cancer. Our findings raise the possibility that dysregulated lipid homeostasis affects Src activity via lipid-mediated Src SA. Thus, targeting lipid–Src interactions could provide a novel approach to modulating activity in cancer.

# Materials and Methods

## Reagents

pFlagN1L11-hSrc and pMycN1L11-Src vectors were described in Aponte et al (2022). pFlagN1L11-hSrc-Src-K5R/K7R/K9R (Src3R) was obtained by mutagenesis using the QuikChange Site-Directed Mutagenesis Kit (Stratagene) using specific oligonucleotides as follows: 5′:CCCTTCACCATGGGTAGCAACAGGAGCAGGCCCAGGGATGCCAGCCAGCGGCGCCGC, and 5′:GCGGCGCCGCTGGCTGGCATCCCTGGGCCTGCTCCTGTTGCTACCCATGGTGAAGGG.

The pMX-pS-CESAR retroviral vector expressing human SRC was described in Naudin et al (2014). pMX-SRC3R was obtained by inserting BglII-MluI blunt-ended fragments (1,687 pb) from pFlagN1L11-hSrc-Src3R into pMX-pS-CESAR opened by XhoI and blunt-ended.

### Antibodies

Anti-Src pY419 (#2101L), anti-Src (clone 327 #ab16885; Abcam), anti-Myc (#2276S), and anti-pTyr clone pY1000 Sepharose Bead Conjugate (PTMScan) were from CST; anti-Flag (M2 antibody; Sigma-Aldrich), anti-tubulin (gift from N. Morin, CRBM, Montpellier, France), anti-pTyr 4G10 (gift from P. Mangeat, CRBM, Montpellier, France), and anti-cst1 (that recognizes Src, Fyn and Yes) were described in Naudin et al (2014); and anti-rabbit IgG-HRP and anti-mouse IgG-HRP were from GE Healthcare.

## Myristoylated protein expression and purification

Myristoylated SNRE and mutated versions were obtained by co-expression of the N-myristoyltransferase (NMT) enzyme along the Src USH3-His$_6$Tag construct in a pETDuet-1 (Novagen) plasmid. The mutations were introduced using QuikChange II XL Site-Directed Mutagenesis Kit (Agilent) and Q5 High-Fidelity PCR Kit (New England BioLabs).

Plasmids were transformed in *E. coli* Rosetta (DE3)pLysS (Novagen). Bacterial cells were grown in LB medium supplemented with chloramphenicol (25 $\mu$g/ml) and ampicillin (100 $\mu$g/ml) at 37°C until an OD$_{600nm}$ of ~0.6. A freshly prepared solution of myristic and palmitic acid (Sigma-Aldrich) (200 $\mu$M final concentration/each) and fatty acid–free BSA (Sigma-Aldrich) (600 $\mu$M final concentration) was added to the cell culture. The lipid–BSA solution was prepared by adding one equivalent of NaOH, heating at 65°C until complete lipid dissolution, and adjusting the final pH to 8. Protein expression was induced with 1 mM of IPTG (NZYTech) and performed for 5 h at 28°C. 1 h after induction start, 6 g/liter of glucose was added to the cell culture. Cells were harvested at 3,993$g$ for 20 min. Bacteria were lysed in buffer containing 20 mM Tris–HCl, 300 mM NaCl, 10 mM imidazole, pH 8, 1% Triton X-100 (Sigma-Aldrich), 1x Protein Inhibitor Cocktail (Sigma-Aldrich), 1 mM PMSF (Sigma-Aldrich), 25 $\mu$g/ml lysozyme (Sigma-Aldrich), and 5 $\mu$g/ml DNase I (Roche). Cells were sonicated on ice and centrifuged at 48,000$g$ for 45 min. Ni-NTA affinity chromatography was performed using a 1 ml Ni-NTA cartridge (GE Healthcare). The protein was eluted with buffer containing 20 mM Tris–HCl, 300 mM NaCl, 10 mM imidazole, 400 mM imidazole, 0.02% Triton X-100 (Sigma-Aldrich) at

pH 8. The final purification step consisted of a size-exclusion chromatography in a Superdex 75 26/60 column (GE Healthcare) using phosphate buffer (50 mM NaH$_2$PO$_4$/Na$_2$HPO$_4$, 150 mM NaCl, 0.2 mM EDTA, pH 7.5). The protein was concentrated with a Vivaspin 20, 5 kD MWCO concentrator (Sigma-Aldrich). The purity of the protein was established by HPLC in a BioSuite pPhenyl 1000RPC 2.0 × 75 mm; 10 $\mu$m column coupled to mass spectrometry (ESI-MS), confirming the complete protein myristoylation. All proteins were flash-frozen and stored at −80°C. Myristoylated SH4 domain peptides (myrSH4) were synthesized by SynPeptide Co., Ltd.

For the AFM experiments, c-Src constructs were myristoylated following an in vitro myristoylation with recombinant NMT produced in *E. coli*. SNRE-3R, myrSH4-UD-SH3-SH2, was expressed in a pET28-b vector, and full-length Src was expressed in a pACYDuet1 together with the Cdc37 chaperone in cells co-expressing GST-tagged YopH tyrosine phosphatase. All constructs were produced as N-terminal His-tagged SUMO fusion that was used for affinity purification and removed by Ulp1 treatment to provide the native N-terminal sequence ready for myristoylation. All synthetic genes were acquired from GenScript.

Plasmids were transformed in *E. coli* Rosetta (DE3)pLysS (Novagen). Bacterial cells were grown in LB medium supplemented with kanamycin (50 $\mu$g/ml) and chloramphenicol (25 $\mu$g/ml) at 37°C until an OD$_{600nm}$ 0.6–0.8 was reached. Expression was induced with 0.5 mM IPTG and left overnight at 30°C. Cells were harvested at 3,993$g$ for 20 min. Bacteria were resuspended in buffer containing 50 mM NaH$_2$PO$_4$/Na$_2$HPO$_4$, 300 mM NaCl, 20 mM imidazole, pH 8, 1 mM PMSF (Sigma-Aldrich), 1 mM benzamidine (Sigma-Aldrich), 25 $\mu$g/ml lysozyme (Sigma-Aldrich), and 5 $\mu$g/ml DNase I (Roche). Cells were sonicated on ice and centrifuged at 48,000$g$ for 30 min. Ni-NTA affinity chromatography was performed using a 5 ml Ni-NTA cartridge (GE Healthcare). The protein was eluted with buffer containing 50 mM NaH$_2$PO$_4$/Na$_2$HPO$_4$, 300 mM NaCl, 400 mM imidazole, pH 8. The eluted sample was buffer-exchanged using size-exclusion chromatography in a Superdex 75 26/60 (GE Healthcare) to a myristoylation buffer (50 mM NaH$_2$PO$_4$/Na$_2$HPO$_4$, 100 mM NaCl, pH 7.3).

Full-length Src was expressed in *E. coli* B834(DE3)-competent cells. Purification included Ni-NTA affinity purification, Ulp1 cleavage, GSTrap HP (Cytiva) to remove residual GST-tagged YopH that coeluted in the previous purification steps, HiTrap Q HP ion exchange, and a final purification with a Superdex 75 26/60 size-exclusion column.

For in vitro myristoylation, constructs and myristoyl coenzyme A (M4414; Sigma-Aldrich) were mixed at a 1:2 ratio. The mixture was supplemented with 2 $\mu$M of His-tagged recombinant human NMT, 1 mM DTT, and 30 mM CHAPS and left at RT for 2 h. Then, the sample was injected into a 1-ml HiTrap Q HP column to remove the detergent and eluted with a salt gradient. NMT was removed with Ni-NTA beads. The myristoylated constructs were buffer-exchanged to PBS (137 mM NaCl, 2.7 mM KCl, 10 mM Na$_2$HPO$_4$, and 1.8 mM KH$_2$PO$_4$) using a Superdex 75 26/60 (GE Healthcare). The myristoylation state of the protein was established by HPLC in a BioSuite pPhenyl 1000RPC 2.0 × 75 mm; 10 $\mu$m column coupled to mass spectrometry (ESI-MS). The protein was concentrated with an Amicon Ultra-15, 10 kD MWCO concentrator (Merck Millipore). All proteins were flash-frozen and stored at −80°C. Freshly prepared and flash-frozen and

thawed samples had the same retention time by size-exclusion chromatography.

## Preparation of liposomes

1,2-Dioleoyl-sn-glycero-3-phosphocholine (DOPC) (TebuBio) and 1,2-dioleoyl-sn-glycero-3-phospho(1'-rac-glycerol) (sodium salt) (DOPG) (Sigma-Aldrich) were dissolved in chloroform. Lipid compositions were DOPC and DOPC:DOPG (2:1 or 4:1) at 4 mM (for SPR) and 2 mM (for AFM). DOPC:DOPA (9:1) at 2 mM was only used for AFM. The organic solvent was evaporated under a nitrogen stream. Lipid films were rehydrated with vortexing in phosphate buffer 50 mM $NaH_2PO_4/Na_2HPO_4$, 150 mM NaCl, 0.2 mM EDTA, pH 7.5 (SPR assays), or 10 mM $NaH_2PO_4/Na_2HPO_4$, 150 mM NaCl, 5 mM $MgSO_4$, pH 7.5 (AFM experiments). Large unilamellar vesicles were prepared by extrusion using Mini-Extruder (Avanti Polar Lipids). The lipid suspension was extruded 15 times through a 100-nm polycarbonate filter. The mean diameter of the liposomes was verified by dynamic light scattering (Zetasizer Nanoseries S, Malvern Instruments). Liposomes were used within 2 d to avoid lipid oxidation.

## Surface plasmon resonance assays

The complete protocol consists of (1) liposome capture, (2) myristoylated protein binding (including reversible and irreversible bound forms), (3) washing of the major monomeric fraction, (4) antibody binding to the remaining, minor persistently bound fraction, and (5) regeneration with detergents to strip off the captured liposomes. The formation of the lipid bilayer and primary c-Src binding produced clear SPR responses. Persistently bound c-Src dimers represent a minor fraction of the steady-state bound protein causing only a weak shift from the baseline (less than 5% of the primary response dominated by binding of c-Src monomers); thus, the response is amplified with the antibody capture so that the relative amount of dimer formed by c-Src variants can be accurately quantified. We used an antibody directed to the $His_6$Tag located at the C terminus to minimize the interference with the membrane-anchoring region. For myristoylated peptides, lacking a $His_6$Tag we used an antibody directed to the Src N-terminal region. The corresponding mutations in synthetic peptides and longer Src constructs showed equivalent effects, confirming the antibody against Src N terminus was not inducing any artifact.

All measurements were carried out in a Biacore T200 instrument (Cytiva) at 25°C. A 2D-carboxymethyl dextran sensor chip (XanTec) was used. All the flow cells, except for the reference, were modified by a covalent attachment of phytosphingosine (TebuBio) to allow the capture of liposomes. An amine-coupling procedure was performed with 1 mM of phytosphingosine in freshly prepared 20 mM sodium acetate buffer at pH 6.7. The running buffer for all experiments consisted of 50 mM $NaH_2PO_4/Na_2HPO_4$, 150 mM NaCl, 0.2 mM EDTA, pH 7.5. DOPC and DOPC:DOPG (2:1) liposomes at 1 mM concentration were coated over the phytosphingosine-containing flow cells by a 20-s injection at 10 μl/min. In case the distinct flow cells contained different types of lipids, the DOPC liposomes were injected into the first available flow cell to avoid anionic lipid migration toward the neutral liposomes through the flow cells. The reference cell and possible uncovered surface in the liposome flow cells were blocked

with 1 mg/ml of BSA at 50 μl/min for 20 s. Mass transport effects were minimized by injecting the myristoylated c-Src variants at 50 μl/min. Protein concentration ranged from 3 μM to 20 μM for the protein density experiments, and it was fixed to 20 μM for the mutant analysis. The myrSH4 peptides were injected at 50 μM. The protein was allowed to associate for 60 s while the dissociation lasted 350 s. The antibody (anti-$His_6$Tag or anti-Src antibodies; both from Santa Cruz Biotechnology) was injected at 1:5 dilution in running buffer for 60 s at 30 μl/min. n ≥ 3 experiments were performed for the c-Src variants, each time in randomized order. The surface was regenerated with two pulses (30 s at 100 μl/min) of isopropanol:50 mM NaOH (2:3) solution followed by a 20-mM CHAPS (Sigma-Aldrich) or 40-mM octyl beta-glucoside (Sigma-Aldrich) pulse. Each cycle started with freshly captured liposomes. All data were double-referenced (reference channel and baseline subtraction, 0 μM concentration) and analyzed using BiaEval 3.1 software (GE Healthcare).

## Data and statistical analysis

The different mutants were compared relatively with WT using the following equation:

$$Relative\ SPR\ response = \frac{\left\{\frac{R_{SA}}{R_{Lipid}}\right\}_{mutant}}{\left\{\frac{R_{SA}}{R_{Lipid}}\right\}_{average\ WT}},$$

where R is the SPR response observed in the corresponding sensorgram. All the statistical analyses were performed using the Python scientific library SciPy (SciPy: Open-Source Scientific Tools for Python, http://www.scipy.org, last accessed 12 Apr 2022). If not otherwise stated, the analysis was performed against the WT c-Src construct. An unpaired two-sample *t* test (two-sided) was conducted.

## AFM experiments

SLBs were obtained by direct fusion of liposomes onto freshly cleaved mica surfaces (mica disks; Ted Pella). 80 μl of DOPC or DOPC:DOPG (4:1) liposomes (2 mM concentration in 10 mM $NaH_2PO_4/Na_2HPO_4$, 150 mM NaCl, 5 mM $MgCl_2$, pH 7.5) was placed onto the mica surface. After 30 min at RT, the samples were rinsed several times with working buffer (10 mM $NaH_2PO_4/Na_2HPO_4$, 150 mM NaCl, pH 7.5) to remove unfused vesicles, keeping the membranes always hydrated. AFM imaging was performed with NanoWizard 3 BioScience AFM (JPK Instruments; Bruker Nano GmbH) at RT and under liquid environment (working buffer). Silicon nitride probes with a nominal spring constant of 0.35 N/m (DNP-S10; Bruker) were used. After having measured the sensitivity (V/m), the cantilever spring constants were individually calibrated using the equipartition theorem (thermal noise routine). Imaging was performed in AC mode. Lipid membrane formation was verified by imaging and performing force–distance curves on different areas of the samples. Then, protein (or myrSH4 peptide) was added to the buffer solution covering the sample, to a final concentration of 100 nM (or 600 nM for the myrSH4 peptide), and imaging was restarted after 4 min. A FRAP control experiment showed that the overall

stability of the blayers, deposited on glass, was not affected by the addition of protein (Fig S7).

### Cell transfection and retroviral infection

HEK293T and HCT116 cells (ATCC) were cultured in DMEM (Invitrogen) supplemented with 10% FBS, 100 U/ml penicillin, and 100 µg/ml streptomycin. Transfections and retroviral infections were done as described in Aponte et al (2022).

### Proximity ligation assay and confocal microscopy

HEK293T cells plated on glass coverslips were transfected with Src-Flag and Src-Myc mutant constructs for 48 h, and subcellular SRC distribution was analyzed after cell fixation (4% PFA, 0.5% Triton, for 20 min) by fluorescence using fluorescence microscopy as described in Aponte et al (2022). Proximity ligation assay (PLA) was performed according to the manufacturing kit protocol (#DUO92101; MERCK). Briefly, HEK293T cells plated on glass coverslips were transfected with Src-Flag and Src-Myc mutants or PEAK2-Myc (as a positive control) (Lecointre et al, 2018) constructs for 48 h and fixed with 4% PFA, 0.5% Triton during 20 min. Glass slides was blocked in Duolink Blocking Solution during 1 h at 37°C and then incubated in primary antibodies rabbit anti-Flag (#F7425, dilution 1/250; Sigma-Aldrich) and mouse anti-Myc (#9B11, dilution 1/500; Cell Signaling) in the Duolink Antibody Diluent during 1 h at RT. After the washing steps, the slides were incubated with the PLUS and MINUS PLA diluted in the Duolink Antibody Diluent (1:5) during 1 h at 37°C. After three washes, the slides were incubated in the ligation solution during 30 min at 37°C and the amplification solution. Finally, after washes the slides were mounted with the Duolink In Situ Mounting Medium with DAPI and observed with an upright fluorescence microscope. Confocal microscopy analysis was described in Aponte et al (2022). Briefly, images were acquired using a Zeiss LSM 980 NLC confocal microscope equipped with a ×63 Plan-Apochromat oil-immersion objective (NA 1.4) and a pinhole set to 1 Airy, controlled by Zen acquisition software (blue edition). The PLA signal was calculated as the PLA surface area per number of cells of the field (30–50 cells). Five to six fields were analyzed per experiment: from n = 3 independent experiments. For the analysis of Src subcellular distribution analysis by confocal microscopy, optical sectioning was set to 1 µm and digital images were further processed using Fiji software and the intensity line plot analysis plug-in (https://fiji.sc/). Line profiles were extracted as a text file and calculated in Excel to manually detect the cell boundary, which was set to 0 and used as the intensity normalization point, and then, normalized data were collected and graphically and statistically analyzed using Prism software (10–15 images per biological replicate, n = 3). For functional assays in cells, soft agar and cell migration assays were described previously (Naudin et al, 2014; Aponte et al, 2022). For colony formation assay in soft agar, 1,500 cells per well were seeded in 12-well plates in 1 ml DMEM containing 10% FCS and 0.33% agar on a layer of 1 ml of the same medium containing 0.7% agar. After 18–21 d, colonies with >50 cells were scored as positive. The cell migration assay was performed as described in Leroy et al (2009) using FluoroBlok invasion chambers (BD Bioscience). 5,000 cells per well were seeded on top in 0% FCS medium and 10% FCS medium at the bottom. After 30 min at 37°C, the cells were labeled with calcein AM (8 µg/ml; Sigma-Aldrich) and migrative cells were photographed using EVOS FL Cell Imaging System. Quantification of the number of invasive cells per well was done with ImageJ software.

## Data Availability

Original data created for the study are available in the Zenodo repository (Mohammad et al, 2025).

## Supplementary Information

## Acknowledgements

We acknowledge the assistance of Roger Martínez in the preparation of some mutants, Javier Carvajal in some SPR measurements, Daniel Bouvard (CRBM, Montpellier) for confocal microscopy, and Montpellier RIO Imaging for imaging analysis. This work has been funded by the Spanish Agencia Estatal de Investigación (PID2019-104914RB-I00, PDC2021-121629-I00, financed by European Union Next Generation Funds, and PID2022-140459OB-I00 and PID2022-139160OB-IOO cofunded by MICIU/AEI/10.13039/501100011033/ and FEDER A way of making Europe), la Ligue Nationale Contre le Cancer (LNCC), Montpellier SIRIC Grant (INCa-DGOSInserm 6045), the Agence Nationale de Recherche (FUZZY-SRC ANR-21-CE13-0011), and CIBER (Consorcio Centro de Investigación Biomédica en Red, CB06/01/008).

### Author Contributions

I-L Mohammad: investigation and formal analysis.
MI Giannotti: formal analysis, investigation, and methodology.
E Fourgous: formal analysis and investigation.
Y Boublik: investigation.
A Fernández: investigation.
A-L Le Roux: formal analysis, supervision, and investigation.
A Sirvent: formal analysis, investigation, and methodology.
M Taulés: supervision and methodology.
S Roche: conceptualization, formal analysis, supervision, funding acquisition, project administration, and writing—review and editing.
M Pons: conceptualization, data curation, formal analysis, supervision, funding acquisition, validation, project administration, and writing—original draft, review, and editing.

### Conflict of Interest Statement

The authors declare that they have no conflict of interest.

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
