## [Reviewer comments · Life Science Alliance]

Life Science Alliance

Lipid-driven SRC self-association modulates its transformation capacity

Irrem-Laareb Mohammad, Marina Giannotti, Elise Fourgous, Yvan Boublik, Alejandro Fernández, Anabel-Lise Le Roux, Audrey Sirvent, Marta Taulés, Serge Roche, and Miquel Pons

DOI: <https://doi.org/10.26508/lsa.202403019>

Corresponding author(s): Miquel Pons, Universitat de Barcelona and Serge Roche, CNRS

Review Timeline:

Submission Date:	2024-08-27
Editorial Decision:	2024-10-04
Revision Received:	2025-02-13
Editorial Decision:	2025-02-26
Revision Received:	2025-02-28
Accepted:	2025-02-28

Transaction Report:

October 4, 2024

Re: Life Science Alliance manuscript #LSA-2024-03019

Prof. Miquel Pons
Universitat de Barcelona
Laboratory of Biomolecular NMR
Baldiri Reixac, 10
Barcelona 8028
Spain

Dear Dr. Pons,

Thank you for submitting your manuscript entitled "Lipid-driven SRC self-association modulates its transformation capacity" to Life Science Alliance. The manuscript was assessed by expert reviewers, whose comments are appended to this letter. We invite you to submit a revised manuscript addressing the Reviewer comments.

Thank you for this interesting contribution to Life Science Alliance. We are looking forward to receiving your revised manuscript.

Sincerely,

B. MANUSCRIPT ORGANIZATION AND FORMATTING:

Reviewer #1 (Comments to the Authors (Required)):

This work combines atomic force microscopy, mutagenesis, and cell-based experiments to convincingly show that the lipid-anchored Src kinase that has Lys-rich, intrinsically disordered segments forms condensates on membrane surfaces. Condensate formation did not require lipid phase separation or a negative charge. Src self-association enhances oncogenic signaling, and mutation of the Lys cluster to Arg modulates Src self-interaction. The authors have also identified key regions of the protein involved in clustering or condensation.

While it is not clear if and how the micron sized condensates observed in synthetic membranes might be formed in the rather complex membranes of the cell, the findings of this work are still highly significant in that they suggest a potential interplay between small protein-lipid clusters and larger-scale condensates regulating cell signaling processes. This has a far-reaching implication considering the large number of lipid-anchored signaling proteins whose membrane targeting motif is an intrinsically disordered region containing a polybasic domain(s). That said, the authors could do a better job of placing their work in a proper context. This may include, for example, clearly articulating the significance of intrinsic disorder and charge-content in protein and nucleic acid condensation, the prevalence or lack thereof of lipid-mediated protein condensation, the potential role of the two-dimensional membrane surface in self-assembly, the relationship between clustering and phase separation. The comparison to KRAS4B could also be expanded, and more papers could be cited including those dealing with IDRs/IDPs in general but especially those concerned with lipid-modified polybasic IDRs (e.g., doi: 10.1016/j.sbi.2024.102869 and references therein).

Reviewer #2 (Comments to the Authors (Required)):

Since Src kinase is pivotal for numerous cellular processes and its deregulation is associated with tumorigenesis, the molecular details of its activity and specificity are currently of primary interest. It seems that understanding membrane-dependent mechanisms that govern Src activation are critical to decipher how the kinase can be over-activated in cancer cells. Thus, the work by Mohammad and collaborators entitled "Lipid-driven SRC self-association modulates its transformation capacity" nicely fits to the current trend and provides some new and interesting details in light of what was already described in literature. Thus, the content of the manuscript could be interesting to the broad scientific community. In general, the concept of the study is clear and reasonable, the data are carefully documented and the whole story is well described. However, there are some issues listed below that the authors should consider before publishing their work.

- The weakest point of the work is related to AFM imaging. Surely, a clear picture of clusters of various size and character emerges from these data. However, the fact that the "...condensates are about 1,5 nm thinner than the surrounding lipid bilayer..." may suggest that the protein may substantially distort the architecture of the lipid bilayer. The same might be the case in all the studied systems, no matter whether thickness differences were observable or not. To exclude a possibility that SLBs become disrupted during experiments, lipid mobility (e.g. via FRAP) should be measured before and after addition of proteins to the studied systems. The described in the manuscript SLBs were prepared on mica, which is not electrostatically neutral and is known to attract both lipids (particularly charged ones, like PG) and proteins. Maybe that's why the authors observed differences between POPC- and POPC/POPG-based systems?

- In regard to the above, the authors claim that they observed protein-lipid clusters, but there is no evidence that indeed lipids cluster together with proteins. Surely the lipids most probably are important for recruiting proteins and forcing them to interact with each other, but without checking diffusivity it is hard to guess which molecules are clustered and which not.

- The authors should more clearly discuss the specificity of Src towards negatively charged lipids. This issue has been already a matter of a number of research, including work done by the authors (e.g. ref. 3). Some reports point at particular lipid classes, like phosphatidic acid (e.g. DOI: 10.1016/j.ydbio.2013.11.006). I think it is critical for understanding the mechanisms proposed in the manuscript but is definitely not sufficiently exposed there. The first question to address is why in the current research PG was chosen? Within the cytoplasmic leaflet of cellular plasma membranes, a broad range of negatively charged lipids can be found.

- Similarly, the preferential electrostatic interactions of arginine residues with negatively charged lipids deserve a more thorough discussion in light of the most recent discoveries (e.g. DOI: 10.1039/d4sm00088a).

- Were the proteins characterized for proper folding (e.g. via circular dichroism) and aggregation (size-exclusion chromatography and/or dynamic light scattering) directly prior to use (i.e. after thawing the frozen aliquots). This is critical to make sure that

clustering occur only at the lipid bilayer. Are the authors sure that in the third step of SPR (l.543) they wash out only monomeric fraction (not e.g. dimers or smaller oligomers)?

- Minor issues include some typos (e.g. "bleu" l. 635, a few missing or doubled spaces, etc.), editorial errors (e.g. supplementary figure legend do not refer to panels, "rpm" is not appropriate unit to describe centrifugation conditions - "rcf" or "g" does a better job).

Point by point response

Reviewer #1 (Comments to the Authors (Required)):

This work combines atomic force microscopy, mutagenesis, and cell-based experiments to convincingly show that the lipid-anchored Src kinase that has Lys-rich, intrinsically disordered segments forms condensates on membrane surfaces. Condensate formation did not require lipid phase separation or a negative charge. Src self-association enhances oncogenic signaling, and mutation of the Lys cluster to Arg modulates Src self-interaction. The authors have also identified key regions of the protein involved in clustering or condensation.

While it is not clear if and how the micron sized condensates observed in synthetic membranes might be formed in the rather complex membranes of the cell, the findings of this work are still highly significant in that they suggest a potential interplay between small protein-lipid clusters and larger-scale condensates regulating cell signaling processes. This has a far-reaching implication considering the large number of lipid-anchored signaling proteins whose membrane targeting motif is an intrinsically disordered region containing a polybasic domain(s).

We thank the reviewer for the recognition of the relevance of this work.

That said, the authors could do a better job of placing their work in a proper context. This may include, for example, clearly articulating the significance of intrinsic disorder and charge-content in protein and nucleic acid condensation, the prevalence or lack thereof of lipid-mediated protein condensation, the potential role of the two-dimensional membrane surface in self-assembly, the relationship between clustering and phase separation.

We thank the reviewer for the valuable suggestions. We have expanded the introduction to place our work in a broader context emphasizing the role of polybasic lipidated motifs in SFK and small GTPases. We have highlighted the similarities and differences between these two families of membrane-anchoring motifs, particularly the presence of a longer intrinsically disordered region in SFK adjacent to the polybasic lipidated unit. The formation of the large condensates observed by AFM requires the presence of the disordered region beyond the SH4 domain. The interplay between clustering, which is already observed in the lipidated SH4 peptide, and condensate formation, which require additional protein regions, suggests that condensate formation may require the formation of clusters with an intermediate strength. When lysine residues are replaced by arginine, small cluster formation is enhanced, but condensate formation is prevented. A similar effect was observed in the presence of phosphatidic acid, which is expected to interact strongly with the lysine cluster.

The comparison to KRAS4B could also be expanded, and more papers could be cited including those dealing with IDRs/IDPs in general but especially those concerned with

lipid-modified polybasic IDRs (e.g., doi: 10.1016/j.sbi.2024.102869 and references therein).

We have included additional references to KRAS4B and other small GTPases.

Reviewer #2 (Comments to the Authors (Required)):

Since Src kinase is pivotal for numerous cellular processes and its deregulation is associated with tumorigenesis, the molecular details of its activity and specificity are currently of primary interest. It seems that understanding membrane-dependent mechanisms that govern Src activation are critical to decipher how the kinase can be over-activated in cancer cells. Thus, the work by Mohammad and collaborators entitled "Lipid-driven SRC self-association modulates its transformation capacity" nicely fits to the current trend and provides some new and interesting details in light of what was already described in literature. Thus, the content of the manuscript could be interesting to the broad scientific community. In general, the concept of the study is clear and reasonable, the data are carefully documented and the whole story is well described.

We thank the reviewer for the appreciation shown for the work presented.

However, there are some issues listed below that the authors should consider before publishing their work.

- The weakest point of the work is related to AFM imaging. Surely, a clear picture of clusters of various size and character emerges from these data. However, the fact that the "...condensates are about 1,5 nm thinner than the surrounding lipid bilayer..." may suggest that the protein may substantially distort the architecture of the lipid bilayer. The same might be the case in all the studied systems, no matter whether thickness differences were observable or not. To exclude a possibility that SLBs become disrupted during experiments, lipid mobility (e.g. via FRAP) should be measured before and after addition of proteins to the studied systems.

We thank the reviewer for their observation. Indeed, condensates and clusters locally affect the lipid bilayer, as observed in the AFM images.

As suggested by the reviewer, we monitored lipid mobility in a supported lipid bilayer using FRAP in an epifluorescence microscope. The recovery of the fluorescence of a labeled lipid probe was comparable before and after the addition of a myristoylated protein confirming the absence of an overall effect on the bilayer structure or lipid mobility. Due to the limited resolution, the observed recovery is dominated by the protein-free regions and therefore mainly reflects the integrity of the lipid membrane after protein incorporation. This result is consistent with the images obtained using AFM.

The described in the manuscript SLBs were prepared on mica, which is not electrostatically neutral and is known to attract both lipids (particularly charged ones, like PG) and proteins. Maybe that's why the authors observed differences between POPC- and POPC/POPG-based systems?

The formation of DOPC/DOPG bilayers on mica was confirmed by measuring several force-separation curves in the absence of protein. During compression, the membrane is elastically deformed and suddenly pierced by the AFM probe at certain force (breakthrough force F_b) that depends on the system and the probe (Redondo-Morata, L., Losada-Pérez, P., Giannotti, M.I. *Current topics in membranes*, 2020, 86, 1-55). This is shown as a clear discontinuity in the approach curve, as exemplified here:

- In regard to the above, the authors claim that they observed protein-lipid clusters, but there is no evidence that indeed lipids cluster together with proteins. Surely the lipids most probably are important for recruiting proteins and forcing them to interact with each other, but without checking diffusivity it is hard to guess which molecules are clustered and which not.

We thank the reviewer for their observation. The exact structure of the condensates remains unknown. The AFM images shown in figures 2 and 3 of the article indicate that they are especially inhomogeneous in the presence of DOPG. An additional image of a condensate formed in the presence of DOPG is provided below.

Negatively charged lipids create a protruding barrier surrounding the condensate, with a height similar to that of small clusters. The structural differences observed in condensates formed with or without negatively charged lipids confirm that lipids are integral components of the condensates. Interestingly, condensates form also in pure DOPC bilayers, where protein-lipid electrostatic interactions are expected to be weaker than in the presence of DOPG.

We also measured force curves inside the condensates formed in DOPC:DOPG and the surrounding area. Examples of force curves (force vs. Z-piezo position) measured at the positions marked by the highlighted pixel in the force map are shown. Breakthrough events are observed both inside and outside of the condensates, supporting the presence of lipids as integral components of the condensates, although inhomogeneity was confirmed.

Within the condensate region, these breakthrough events appeared more irregular, with some possibly corresponding to thicker structures or even vesicles or more disordered films.

These results have been included as supplementary information.

The first question to address is why in the current research PG was chosen? Within the cytoplasmic leaflet of cellular plasma membranes, a broad range of negatively charged lipids can be found.

The interplay between lipid-phase separation and the formation of protein condensates and clusters is complex. Reports on the incorporation of SFK in lipid rafts often assume that the process is driven by the preference of the protein for a preexisting separate lipid phase.

In this work we want to emphasize the intrinsic capacity of Src N-terminal region to form condensates, independently of any potential phase separation of the lipids to which they are anchored. For this reason, we focused on either a pure, zwitterionic lipid (DOPC) or DOPC:DOPG mixtures that are known not to form separate phases.

- The authors should more clearly discuss the specificity of Src towards negatively charged lipids. This issue has been already a matter of a number of research, including work done by the authors (e.g. ref. 3). Some reports point at particular lipid classes, like phosphatidic acid (e.g. DOI: 10.1016/j.ydbio.2013.11.006). I think it is critical for understanding the mechanisms proposed in the manuscript but is definitely not sufficiently exposed there.

We agree with the reviewer that the specificity of the polybasic lipidated motif of Src for different lipid classes, as previously observed for small GTPases, may play an important role in regulating condensate formation and influencing protein activity. However, incorporating the full complexity of multiple lipid compositions was beyond the scope of this article.

Following the reviewer's suggestion, we prepared phosphatidic acid- containing supported bilayers (DOPC with 10% DOPA). The addition of myristoylated protein led to the formation of clusters, like those observed in the mutant form where lysine residues were replaced with arginine. However, no condensates were formed. This suggests that increasing the strength of the interaction between negatively charged lipids and the polybasic lipidated motif favors the formation of lipid-protein clusters but prevents the formation of condensates involving the adjacent disordered region.

These observations imply that changes in lipid composition, such as increasing the concentration of phosphatidic acid (or other highly charged lipids) may destabilize the condensates and favor the formation of small clusters. The data from Brad Stith showing the increase in activity of Src following an increase in phosphatidic acid aligns with our observation that replacing lysine by arginine in Src results in condensate destabilization, increased cluster formation and enhanced transforming activity.

While these results are promising and open new avenues for exploration, they should be considered preliminary, as the complexity of the PC:PA system is significantly higher than that of the PC:PG system we initially selected to avoid lipid segregation and potential protein partitioning between lipid phases. Further studies fall beyond the scope of the present manuscript, and we are presenting these results as supplementary material.

DOPC:DOPA (9:1)

No protein

+ SNRE

- Similarly, the preferential electrostatic interactions of arginine residues with negatively charged lipids deserve a more thorough discussion in light of the most recent discoveries (e.g. DOI: 10.1039/d4sm00088a).

The well documented preferential interaction of arginine versus lysine residues with negatively charged lipids motivated the study of the R3 mutant. We have included additional references to experimental and computational studies that highlight this important difference. Interestingly, arginine does not simply provide a stronger interaction but also a more selective one, with respect to anionic lipid types.

In the case of Src it seems that nature has selected not for the stronger interaction (with arginine) but for a weaker interaction via a lysine cluster, which facilitates the modulation of self-association by altering the lipid composition. This presents an exciting hypothesis that warrants further investigation, although it extends beyond the scope of the present article.

- Were the proteins characterized for proper folding (e.g. via circular dichroism) and aggregation (size-exclusion chromatography and/or dynamic light scattering) directly prior to use (i.e. after thawing the frozen aliquots). This is critical to make sure that clustering occur only at the lipid bilayer.

We specifically checked by size exclusion chromatography (figure attached) that myristoylated SNRE could be frozen and thawed without any effect. This control has been mentioned in the experimental section.

In a previous study (iScience 2019,12: 194. doi:10.1016/j.isci.2019.01.010.) we measured the NMR spectra of myristoylated and non-myristoylated forms of SNRE. The NMR spectra outside the lipid binding region, showed no major changes in the lipid-bound and non-myristoylated protein confirming that lipid binding does not change the structure of the major form observed by NMR. Additionally, in another study, we confirmed that self-association observed using single-molecule fluorescence of GFP fused to myristoylated SH4 and Unique domain, when deposited on lipids, was not observed when the myristoylated protein was deposited on polylysine-coated glass coverslips (Chemistry Select 2016, 4, 642 DOI: 10.1002/slct.201600117)

Are the authors sure that in the third step of SPR (1.543) they wash out only monomeric fraction (not e.g. dimers or smaller oligomers)?

The SPR results operatively distinguish between reversible and persistent binding. Any form that remains attached to the SPR chip after washing is classified as “persistently bound”. This category may include dimers, larger oligomers or condensates.

Our previous work on the binding kinetics of the “persistently bound” form indicated a second-order concentration effect, suggesting the presence of, at least, dimers. Single-molecule characterization of an extensively washed SLB showed the presence of dimers and trimers in the remaining protein. We also tested their dissociation with increasing washing times and the fraction of “persistently bound” form did not decrease significantly after 2 hours of washing. So, while we cannot entirely rule out that a small fraction of dimers is retained during the washing, our results strongly suggests that any

self-associated species (dimers, trimers or higher oligomers) are retained and detected by the antibody in the SPR experiments.

- Minor issues include some typos (e.g. "bleu" l. 635, a few missing or doubled spaces, etc.), editorial errors (e.g. supplementary figure legend do not refer to panels, "rpm" is not appropriate unit to describe centrifugation conditions - "rcf" or "g" does a better job).

We have indicated the g values that correspond to the centrifugation conditions with the used rotors. Bleu is the denomination of the software used by the French group. We have tried to correct additional typing or grammatical mistakes. We apologize for the errors.

February 26, 2025

RE: Life Science Alliance Manuscript #LSA-2024-03019R

Prof. Miquel Pons
Universitat de Barcelona
Laboratory of Biomolecular NMR
Baldiri Reixac, 10
Barcelona 8028
Spain

Dear Dr. Pons,

Thank you for submitting your revised manuscript entitled "Lipid-driven SRC self-association modulates its transformation capacity". We would be happy to publish your paper in Life Science Alliance pending final revisions necessary to meet our formatting guidelines.

- please address Reviewer 2's remaining comments
- please be sure that the authorship listing and order is correct
- please add the X and Buleasky handles of your host institute/organization as well as your own or/and one of the authors in our system
- please add Author Contributions to our system as well
- please add your main, supplementary figure, and table legends to the main manuscript text after the references section
- please consult our manuscript preparation guidelines <https://www.life-science-alliance.org/manuscript-prep> and make sure your manuscript sections are in the correct order
- please add callouts for Figure S3A-B to your main manuscript text
- please update the Data Availability statement to include links to the datasets

FIGURE CHECK:

- please add sizes next to all blots
- please add a scale bar to Figure 5B

A. FINAL FILES:

B. MANUSCRIPT ORGANIZATION AND FORMATTING:

Thank you for your attention to these final processing requirements. Please revise and format the manuscript and upload materials within 5 days.

Sincerely,

Reviewer #2 (Comments to the Authors (Required)):

I would like to thank the authors for addressing all the issues raised by the reviewers. The manuscript improved a lot and is now ready for publication. There are only two minor suggestions from my side:

- I would include FRAP results as a supplementary information.
- Regarding ZEN software, "blue edition" is what was officially released by ZEISS.

February 28, 2025

RE: Life Science Alliance Manuscript #LSA-2024-03019RR

Prof. Miquel Pons
Universitat de Barcelona
Laboratory of Biomolecular NMR
Baldiri Reixac, 10
Barcelona 8028
Spain

Dear Dr. Pons,

Thank you for submitting your Research Article entitled "Lipid-driven SRC self-association modulates its transformation capacity". It is a pleasure to let you know that your manuscript is now accepted for publication in Life Science Alliance. Congratulations on this interesting work.

DISTRIBUTION OF MATERIALS:

Again, congratulations on a very nice paper. I hope you found the review process to be constructive and are pleased with how the manuscript was handled editorially. We look forward to future exciting submissions from your lab.

Sincerely,
